# Effectiveness of Crohn’s Disease Exclusion Diet for Induction of Remission in Crohn’s Disease Adult Patients

**DOI:** 10.3390/nu13114112

**Published:** 2021-11-17

**Authors:** Martyna Szczubełek, Karolina Pomorska, Monika Korólczyk-Kowalczyk, Konrad Lewandowski, Magdalena Kaniewska, Grażyna Rydzewska

**Affiliations:** 1Clinical Department of Internal Medicine and Gastroenterology with Inflammatory Bowel Disease Subunit, Central Clinical Hospital of Ministry of the Interior and Administration, 02-507 Warsaw, Poland; k.m.pomorska@gmail.com (K.P.); monika.korolczyk@o2.pl (M.K.-K.); dr.k.lewandowski@icloud.com (K.L.); gastroenterologia@cskmswia.gov.pl (M.K.); grazyna.rydzewska@cskmswia.gov.pl (G.R.); 2Collegium Medicum, Jan Kochanowski University, 25-317 Kielce, Poland

**Keywords:** Crohn’s disease, Crohn’s disease exclusion diet, dietary therapy

## Abstract

Exclusive enteral nutrition (EEN) is a first-line treatment in active, mild to moderate Crohn’s disease (CD) in children. The Crohn’s disease exclusion diet (CDED), which avoids products known to have a pro-inflammatory effect on the intestinal mucosa, presents similar effectiveness to EEN for inducing remission in the paediatric population. The aim of the study was to evaluate the effectiveness of the CDED in inducing remission in adult patients. Between March 2020 and May 2021, 32 patients in a gastroenterology outpatient centre were treated according to the assumptions of the CDED. The patients were seen at baseline, at week 6, and at week 12 of the study. During the visits, anthropometric measurements and laboratory tests were performed, Crohn’s disease activity index (CDAI) was calculated, and the Inflammatory Bowel Disease Questionnaire (IBDQ) was completed. The study included a total of 32 participants, 18 women (56.3%) and 14 men (43.7%). Clinical remission was obtained in 76.7% patients after 6 weeks and in 82.1% after 12 weeks of therapy. Calprotectin levels were significantly lower in the second follow-up compared with baseline (*p* = 0.021). The CDED is an effective therapy for inducing remission in the adult CD population.

## 1. Introduction

Inflammatory bowel disease (IBD) includes Crohn’s disease (CD) and ulcerative colitis (UC). Despite the long history of research into IBD, its aetiology is still not fully understood [1]. According to current knowledge, environmental, genetic and immune factors are considered the major causes of IBD [2]. We are currently observing a much more significant role of environmental factors in the development of IBD than previously assumed. Genetic variants are responsible for approximately 26% of CD and 19% of UC cases [3].

Environmental factors involved in the pathogenesis of IBD influence the intestinal microbiota, its penetration into colon mucous, and effects on intestinal permeability. In animal models that spontaneously develop colitis and in patients with active IBD, bacteria penetrate the colon mucus and reach the epithelium [4]. Bacterial invasion into the mucosa was observed in 83.3% of colonic specimens from UC patients and in up to 55.6% of specimens from CD patients but was not present in the tissues of healthy individuals [5]. Inflammation of the intestinal membrane can be promoted by the loss of those microorganisms that are known to have a beneficial role in intestinal homeostasis [6].

Diet is one of the key players in maintaining a normal gut microenvironment. It has a great impact on the development and treatment of IBD. A widely used dietary intervention recommended by ECCO/ESPGHAN as the first-line treatment to induce remission in new-onset, mild to moderate CD in children is exclusive enteral nutrition (EEN). It is based on a supply of protein formula for 6–8 weeks that covers 100% of daily caloric intake [7]. The effectiveness of EEN is approximately 80%, which is comparable to corticosteroid therapy [8,9,10]. There have been attempts to use EEN in adult CD patients. The results of one study that compared EEN with corticosteroids were similar for adults who completed the treatment to the child population [11]. Unfortunately, non-compliance with EEN intervention can reach 50% of participants due to poor motivation, a lack of support, and an unpalatable formula [11,12]. Diets that include regular food products could be better tolerated by patients. Studies have shown that partial enteral nutrition (PEN) with free diet is ineffective in inducing remission in CD. The effectiveness of PEN vs. EEN was 15% vs. 42%, respectively [13]. It was suspected that the lack of presence of products with a proven negative effect on the intestines, not the enteral nutrition itself, is responsible for mucosal healing. According to the statement, Professor Arie Levine and colleagues developed a dietary intervention for active CD in children and young adults called the Crohn’s disease exclusion diet (CDED). Between 2011 and 2013, 47 patients followed instructions regarding their diet for 12 weeks, with a special focus on avoiding the products disallowed because of their negative effect on the intestinal mucosa. A response was achieved in 78.7% of the participants, and 70.6% of the patients went into remission [14].

The aim of our study was to evaluate the effectiveness of CDED for inducing remission in adult patients with active CD.

## 2. Materials and Methods

This is a report of our experience with CDED in adult patients with active CD. Between March 2020 and May 2021, 32 patients in a gastroenterology outpatient centre were treated according to the assumptions of the CDED. CD patients with a calculated Crohn’s disease activity index (CDAI) value of >150 points were included in the study. Patients undergoing pharmacological treatment with 5-ASA (mesalazine or sulfasalazine), immunosuppressants (azathioprine or methotrexate), or biological treatment in the maintenance phase who lost treatment response (increase in CDAI >150 points) were accepted in the study. The pharmacological treatment must have lasted for more than 8 weeks, in fixed doses, to exclude the effect of pharmacological treatment modification for the outcome of CDED. Patients who did not take the voluntary and informed decision to participate in the study; those with a known intolerance or hypersensitivity to the components of Modulen (Nestlé, Vevey, Switzerland); those undergoing treatment with glucocorticosteroids or antibiotics; those with a coexisting infection of the gastrointestinal tract determined by a positive stool culture; those whose 5-ASA, immunosuppresants, or biological drug treatment lasting less than 8 weeks; and those with fistulas, abscesses, or stomas were excluded from the study.

The patients were seen at baseline, at week 6, and at week 12 of the study. During the visits anthropometric measurements (height, weight, and BMI) and laboratory tests (peripheral blood count, CRP, ESR, ALT, AST, total bilirubin, ALP, GGTP, creatinine, urea, uric acid, sodium, potassium, calcium, iron, ferritin, vitamin B12, folic acid, vitamin D3, albumin, total protein, calprotectin, and a faecal culture) were performed, CDAI was calculated, and the Inflammatory Bowel Disease Questionnaire (IBDQ) was filled in (to evaluate the quality of life). All patients received instructions regarding the diet and list of food products assigned to one of three categories—mandatory food (patients were obligated to consume all food products from the list), allowed foods and beverages (list of products with proven no negative effect on intestinal mucosa), disallowed foods and beverages (list of products with pro-inflammatory effect on intestine). The first 6-week period involved a restricted diet, where 50% of the calculated energy intake was supplied with mandatory food (150–200 g of chicken breast, two eggs, two potatoes, two bananas, and one apple per day) and integrated with the allowed foods to prepare daily meals. To prevent malnutrition, the other 50% of caloric intake was delivered with a complete, liquid formula (Modulen, Nestlé, Vevey, Switzerland). Individuals who refused to drink the calculated amount of formula were able to follow the diet without or with smaller portions of additional supplementation. Dairy products, animal fats, emulsifiers, and processed, canned, industrially frozen, or dried food was not allowed. In the second 6-week period, the supply of formula was reduced to 25% of daily caloric intake. The number of allowed food products was increased, the meals became more varied, and the dietary restrictions were less demanding. No specific diet was implemented after week 12. A dietician gave instructions on healthy eating habits and avoiding highly processed foods.

Our primary endpoint was the achievement of remission at week 12, defined as a CDAI of less than 150 points. The secondary endpoints were the inducement of a clinical response, understood as a reduction in CDAI by 100 points or more; a statistically significant drop in calprotectin level and inflammatory markers (CRP and WBC); improved quality of life, evaluated by the IBDQ; and an improvement in laboratory parameters (albumin, total protein, vitamin D3, vitamin B12, folic acid, sodium, potassium, calcium, iron, and ferritin levels) and BMI, according to the reference values.

Data analysis was based on an intention-to-treat approach. Statistical analysis was carried out using R software, version 4.0.5 (http://cran.r-project.org, accessed on 27 September 2021). Nominal variables are presented as count *n* (% frequency) and continuous variables as mean ± SD or median (Q1; Q3—lower quartile; upper quartile). Normality of the distribution was validated using the Shapiro–Wilk test, skewness, and kurtosis values and on visual assessment of histograms. Comparison between three measurements (baseline and first and second follow-up) was conducted with ANOVA for repeated measures or a series of Wilcoxon signed-rank tests. Bonferroni correction for multiple comparisons was applied. Variance sphericity was tested with the Mauchly test and Greenhouse–Geisser and Huynh–Feldt corrections for departure from sphericity were applied as appropriate. The McNemar test was used to compare CDAI level (understood as light/moderate/severe active CD) between three measurements, but due to low populations of subgroups, comparisons between the second follow-up and baseline and between the second and first follow-up were not possible. Additionally, 95% binomial confidence intervals (CI) for proportions were calculated. All tests were based on α = 0.05. Power of sample size was determined post-hoc using G*Power 3.1.9.2 software. For determining that percentage of patients who achieved clinical remission (CDAI < 150) was significantly different from 0, and assuming alpha level = 0.05, in the first follow-up as well as second follow-up, we achieved the power level (1 minus beta) of 1.0.

## 3. Results

The study included a total of 32 participants, 18 women (56.3%) and 14 men (43.7%). Four patients (12.5%) gave initial consent to follow the CDED but refused to continue dietary restrictions before the end of week 12.

The average age in the group was 31.42 ± 9.01 years, ranging from 20 to 62 years. The BMI at baseline was 21.40 kg/m2 (19.15; 24.73), ranging from 14.50 to 35.30. The baseline CDAI was 253 points (175.50; 373.00) (from 151 to 562 points). Thirteen patients (40.6%) suffered from mild CD, defined as a CDAI of 150–220 points; 15 (46.9%) represented moderate activity of CD (a CDAI of 221–450 points); and 4 (12.5%) participants had severe CD (a CDAI of >450 points). Baseline calprotectin was at a median level of 393.00 µg/g (58.85; 969.00), ranging from 11.9 to 4 630 µg/g. A summary of other baseline parameters is presented in Table 1.

A CDAI level of less than 150 points, known as clinical remission, was present after six weeks of treatment vs baseline in 76.7% of patients (CI95 (57.7–90.1)) and in 82.1% of participants (CI95 (63.1–93.9)) after 12 weeks of treatment vs baseline.

A clinical response, understood as a decline in the CDAI level of more than 100 points in comparison to baseline, was noted in 83.3% of the patients (CI95 (65.3–94.4%)) after six weeks of treatment and in 85.7% of cases (CI95 (67.3–96.0%)) after 12 weeks of treatment. The data are presented in Figure 1 and Figure 2.

Analysis of changes in follow-up measurements is presented in Table 2 and Figure 3. A statistically significant drop in CDAI was observed after six weeks (*p* < 0.001) and after 12 weeks (*p* < 0.001) of dietary intervention as compared to baseline. IBDQ also noted significant changes between measurements (*p* < 0.001) with significantly higher levels at the first and second follow-up vs baseline. Calprotectin levels were significantly lower at the second follow-up vs baseline (*p* = 0.021).

While at baseline, almost half of the patients had moderate CD (46.9%, CI95 (29.1–65.3%)), at the first follow-up, it was 10.0% (CI95 (2.1–26.5%)); at the second follow-up, it was only one patient (3.6%, CI95 (0.1–18.4%)). A CDAI of up to 220 points was present in 40.6% of cases (CI95 (23.7–59.4%)) at baseline, in 90.0% (CI95 (73.5–97.9%) at the first follow-up, and in 92.9% at the second follow-up (CI95 (76.5–99.1%)). Severe CD was present in four patients at baseline, no patients at the first follow-up, and one patient at the second follow-up (Figure 4).

## 4. Discussion

EEN in CD has a strong and well-documented effectiveness in the child population. ECCO/ESPGHAN recommends it as the first-line treatment to induce remission in new-onset, mild to moderate CD [7]. In various studies, remission was achieved in up to 85% of patients using EEN [8,9,15]. Encouraging results led to the first attempts of this nutritional therapy in adults. However, non-adherence to the treatment contributed to limited success of EEN in many studies. The poor cooperation of the adults resulted from little motivation to complete the treatment, a lack of support, and the disagreeable taste of the formula [11].

Including meals based on regular food products in the diet helped to achieve better tolerance of the dietary regime. The idea of the CDED is to exclude products known to have a pro-inflammatory effect on the intestinal mucosa. In the publication by Levine et al., the effectiveness of a 12-week therapy with the CDED and PEN was compared with a six-week treatment with EEN and a gradual return to a free diet afterwards. Clinical remission at week six was similar in the two groups. In the group with a return to a free diet, after being exposed to all available food products, calprotectin levels started to increase. In contrast, calprotectin continued to fall for the next six weeks in the CDED group [14].

In a study by Johnson et al., using PEN with a free diet resulted in CD remission in 15% of patients [13]. These reports confirm the assumption that the lack of presence of certain harmful food products, not the EEN, might be responsible for reducing intestinal inflammation.

Current data indicate a similar effectiveness of the CDED combined with PEN in comparison to EEN in children for inducing remission in CD. In a study by Sigall-Boneh et al., a response to the CDED with PEN was obtained in 78.7% of children and young adults, while remission was obtained in 70.6% [16]. In a randomised, controlled trial by Levine et al., 85% of the patients on the CDED with PEN demonstrated a clinical response, and 80% entered clinical remission [14]. The above data correspond with the results obtained in our study, in which clinical remission was observed in 76.7% of patients after six weeks and 82.1% after 12 weeks of treatment. A clinical response was observed in 83.3% and 85.7% of participants after six and 12 weeks of treatment, respectively. The difference between our study and those mentioned above concerns the target population. Our study provides evidence for similar effectiveness with the CDED in children and adults.

Only four of the 32 patients failed to comply with the dietary recommendations, which is markedly different to the EEN, where almost half of the participants terminated the treatment due to non-compliance [11].

In studies involving dietary therapy, the study populations were primarily children and young adults with a short duration of disease and mild to moderate disease activity [8,9,15,16]. In adults, over the years, the disease becomes more advanced and results in complications. Despite this, we did not observe a lower response rate to dietary treatment in our study.

In addition, we included patients who were not treatment-naïve; some of them had participated in biological treatment programmes but failed to achieve clinical remission. Four participants suffered from severe CD, determined by a CDAI of over 450 points. Three of four patients achieved clinical remission (a CDAI of <150) after 12 weeks of CDED treatment. In a study by Sigall-Boneh et al., dietary intervention resulted in success in three out of five patients who failed to achieve remission or who no longer responded to biological treatment [16]. This sheds new light on the selection of patients qualifying for dietary treatment. Perhaps disease severity should not be an exclusion criterion for an intervention.

An objective indicator of reduction of inflammation in the intestines is faecal calprotectin [17]. It is a useful tool that provides evidence of mucosal healing [18]. Its value cannot be influenced by subjective factors, like general well-being, which are taken into account in CDAI. The level of faecal calprotectin decreased significantly during our therapeutic intervention.

We decided to validate the effect of CDED treatment on health-related quality of life in patients with CD. After both six and 12 weeks of observations, we noted an increase in the IBDQ value. The treatment resulted in direct benefits and improvement not only in laboratory test results but also in the quality of life of our patients.

We did not observe any significant changes in BMI level, albumin, or protein concentration before and after therapeutic intervention. Although the CDED is an elimination diet, it did not lead to significant weight loss or malnutrition in our patients. Moreover, we noted an increase in the concentration of vitamins, iron, and calcium after six and 12 weeks of therapy.

The main limitation of the study is the observational, open-label character of trial without control group. During the COVID-19 epidemic, attempts are being made to avoid weakening immune response, which is associated with the use of glucocorticoids. Almost complete lack of side effects regarding nutritional treatment encouraged us to apply CDED to all consenting patients.

Another limitation is that we did not directly assess mucosal healing by endoscopy. Achievement of remission was supported by the results of biomarkers, such as calprotectin.

Despite limitations of the trial, several aspects contribute to the novelty of this report. It is the first known study evaluating the effects of CDED for induction of remission in the adults. However, a randomized trial is needed to obtain higher-level evidence.

## Figures and Tables

**Figure 1 nutrients-13-04112-f001:**
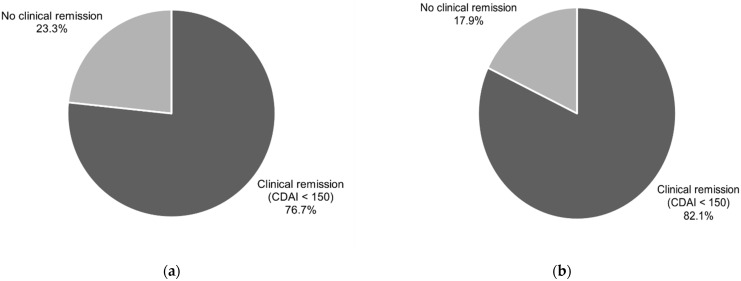
Clinical remission (CDAI < 150) in patients after 6 (**a**) and after 12 (**b**) weeks of CDED.

**Figure 2 nutrients-13-04112-f002:**
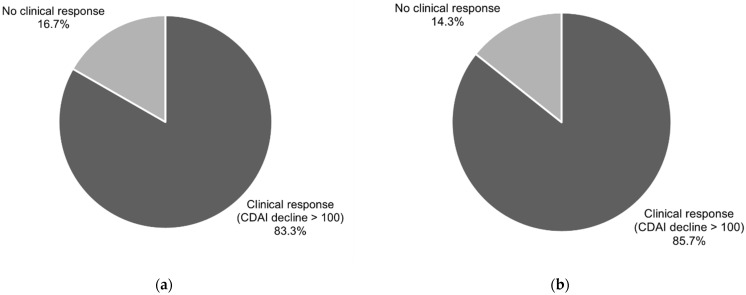
Clinical response (reduction in CDAI > 100 points) in patients after 6 (**a**) and after 12 (**b**) weeks of CDED.

**Figure 3 nutrients-13-04112-f003:**
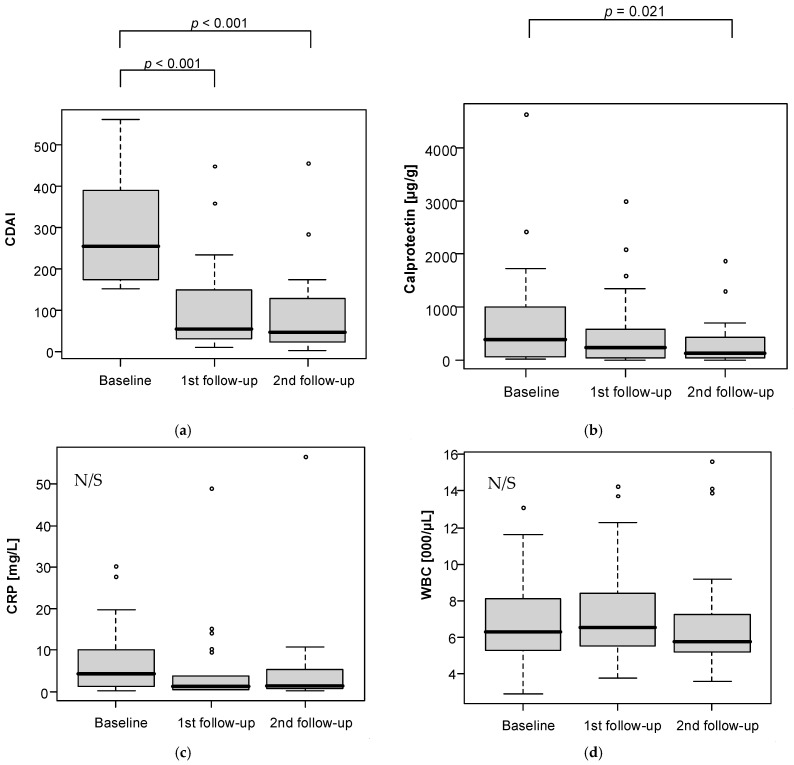
Analysis of changes in follow-up measurements regarding: (**a**) CDAI level; (**b**) calprotectin level; (**c**) CRP level; (**d**) WBC level; (**e**) IBDQ level.

**Figure 4 nutrients-13-04112-f004:**
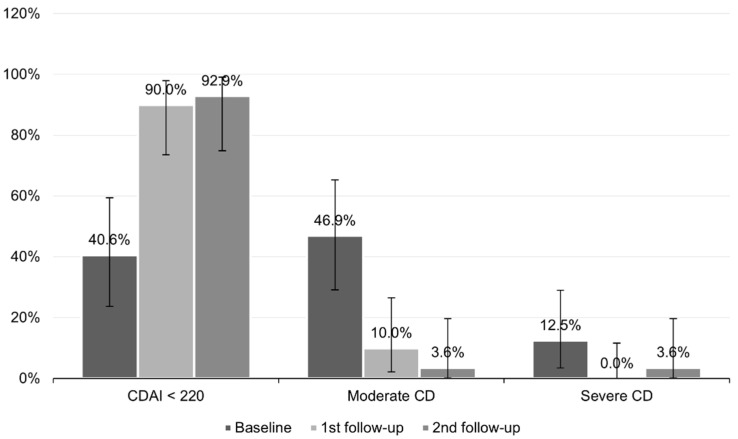
Severity of CD in patients at baseline, 1st follow-up, and 2nd follow-up. Horizontal lines represent 95% confidence intervals (CI).

**Table 1 nutrients-13-04112-t001:** Baseline characteristics.

	Level
N	32
Sex, female, *n* (%)	18 (56.3)
Age, years	31.42 ± 9.01
Height, m	1.73 ± 0.10
Weight, kg	67.31 ± 16.44
BMI, kg/m^2^	21.40 (19.15; 24.73)
CDAI, median (Q1; Q3)	253.00 (175.50; 373.00)
Mean ± SD	290.79 ± 127.04
IBDQ	125.44 ± 36.61
Calprotectin, median (Q1; Q3), µg/g	393.00 (58.85; 969.00)
Mean ± SD	693.56 ± 940.49
WBC, 10^3^/µL	6.76 ± 2.31
RBC, 10^6^/µL	4.67 ± 0.61
PLT, 10^3^/µL	303.10 ± 91.27
Hgb, g/dL	13.32 ± 1.82
Ht, %	40.13 ± 4.31
ESR, mm/h	9.00 (2.75; 22.25)
CRP, mg/L	4.30 (1.10; 10.05)
AlAT, U/L	17.00 (11.00; 31.50)
AspAT, U/L	19.00 (15.50; 25.00)
Bilirubin, mg/dL	0.49 ± 0.26
ALP, U/L	79.54 ± 31.93
GGTP, U/L	14.00 (10.00; 26.00)
Creatinine, mg/dL	0.79 ± 0.12
Urea, mg/dL	25.77 ± 5.78
Uric acid, mg/dL	5.10 (3.90; 6.20)
Na, mmol/L	140.90 ± 1.94
K, mmol/L	4.35 ± 0.32
Ca, mmol/L	2.34 ± 0.12
Fe, µg/dL	74.94 ± 45.63
Ferritin, ng/mL	61.00 (22.50; 154.00)
Vit. B12, pg/mL	407.56 ± 224.08
Folic acid, ng/mL	5.80 (3.88; 12.40)
Albumin, g/dL	4.37 ± 0.51
Total protein, g/dL	7.24 ± 0.68
Vit. D3, ng/mL	26.31 ± 12.28

Data presented as mean ± SD or median (Q1; Q3) unless otherwise indicated.

**Table 2 nutrients-13-04112-t002:** Analysis of changes in follow-up measurements.

	Baseline (*n* = 32)	1st Follow-Up (*n* = 30)	2nd Follow-Up (*n* = 28)	*p* ^1^	Post-Hoc ^2^		
					1st vs. Base	2nd vs. Base	2nd vs. 1st
Weight, kg ^3^	67.31 ± 16.44	67.94 ± 16.31	67.16 ± 14.86	0.622			
BMI, kg/m^2^	21.40 (19.15; 24.73)	21.50 (19.78; 25.40)	21.20 (19.78; 24.35)		>0.999	0.475	0.149
CDAI	253.00 (175.50; 373.00)	53.40 (31.38; 143.00)	45.00 (24.75; 122.00)		<0.001	<0.001	0.339
IBDQ ^3^	125.44 ± 36.61	172.27 ± 34.96	177.57 ± 31.24	<0.001	<0.001	<0.001	>0.999
Calprotectin, µg/g	393.00 (58.85; 969.00)	231.00 (37.20; 577.50)	122.00 (33.98; 387.75)		0.509	0.021	>0.999
WBC, 10^3^/µL ^3^	6.76 ± 2.31	7.26 ± 2.75	6.81 ± 3.00	0.543			
RBC, 10^6^/µL ^3^	4.67 ± 0.61	4.78 ± 0.52	4.84 ± 0.53	0.282			
PLT, 10^3^/µL ^3^	303.10 ± 91.27	301.60 ± 89.23	281.29 ± 75.66	0.437			
Hgb, g/dL ^3^	13.32 ± 1.82	13.57 ± 1.54	13.54 ± 1.37	0.206			
Ht, % ^3^	40.13 ± 4.31	41.14 ± 3.60	41.11 ± 3.28	0.112			
ESR, mm/hr	9.00 (2.75; 22.25)	10.00 (6.00; 16.50)	6.00 (4.00; 16.50)		>0.999	>0.999	>0.999
CRP, mg/L	4.30 (1.10; 10.05)	1.30 (0.50; 3.70)	1.35 (0.60; 4.80)		0.053	0.158	0.920
AlAT, U/L	17.00 (11.00; 31.50)	23.00 (11.50; 31.50)	18.50 (14.50; 30.00)		>0.999	>0.999	>0.999
AspAT, U/L	19.00 (15.50; 25.00)	21.50 (17.00; 24.75)	19.00 (16.75; 22.25)		0.773	>0.999	0.805
Bilirubin, mg/dl ^3^	0.49 ± 0.26	0.52 ± 0.24	0.55 ± 0.36	0.458			
ALP, U/L ^3^	79.54 ± 31.93	70.78 ± 21.55	72.44 ± 22.12	0.165			
GGTP, U/L	14.00 (10.00; 26.00)	14.00 (9.00; 25.00)	14.00 (9.00; 18.50)		0.689	>0.999	>0.999
Creatinine, mg/dL ^3^	0.79 ± 0.12	0.83 ± 0.15	0.82 ± 0.15	0.270			
Urea, mg/dL ^3^	25.77 ± 5.78	29.86 ± 7.86	28.96 ± 9.90	0.104			
Uric acid, mg/dL	5.10 (3.90; 6.20)	5.05 (3.98; 6.05)	4.90 (4.15; 6.05)		>0.999	>0.999	0.857
Na, mmol/L ^3^	140.90 ± 1.94	140.33 ± 2.01	140.82 ± 1.42	0.582			
K, mmol/L ^3^	4.35 ± 0.32	4.43 ± 0.23	4.34 ± 0.33	0.424			
Ca, mmol/L ^3^	2.34 ± 0.12	2.41 ± 0.12	2.39 ± 0.14	0.001	0.002	0.575	0.627
Fe, µg/dL ^3^	74.94 ± 45.63	78.20 ± 38.01	79.48 ± 40.66	0.742			
Ferritin, ng/mL	61.00 (22.50; 154.00)	57.00 (24.50; 113.25)	32.00 (12.50; 80.50)		0.181	0.092	>0.999
Vit. B12, pg/mL ^3^	407.56 ± 224.08	528.59 ± 262.77	488.78 ± 249.55	<0.001	<0.001	0.012	0.002
Folic acid, ng/mL	5.80 (3.88; 12.40)	13.30 (10.30; 15.70)	11.65 (9.25; 16.30)		0.010	0.215	0.552
Albumin, g/dL ^3^	4.37 ± 0.51	4.42 ± 0.49	4.32 ± 0.63	0.259			
Total protein, g/dL ^3^	7.24 ± 0.68	7.27 ± 0.77	7.07 ± 0.87	0.132			
Vit. D3, ng/mL ^3^	26.31 ± 12.28	30.69 ± 12.86	28.58 ± 9.20	0.034	0.025	>0.999	0.053

Data presented as mean ± SD ^3^ or median (Q1; Q3) in remaining cases. Baseline and follow-up measurements compared with ANOVA for repeated measures with paired *t*-test post-hoc analysis ^3^ or with Wilcoxon signed-rank tests in remaining cases. In post-hoc analysis, Bonferroni correction for multiple comparisons was applied; ^1^ result of ANOVA for repeated measures analysis; ^2^ paired *t*-test post-hoc analysis for ANOVA or Wilcoxon signed-rank tests analysis; ^3^ variables analysed with ANOVA for repeated measures analysis.

## Data Availability

Data supporting reported results are available upon request.

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
