# Peer review of "Effectiveness of Crohn’s Disease Exclusion Diet for Induction of Remission in Crohn’s Disease Adult Patients"

_nutrients, 2021, doi:10.3390/nu13114112_

Round 1
Reviewer 1 Report
The authors illustrate a small size study to evaluate the effectiveness of Crohn’s disease exclusion diet (CDED). My concern is about the statistical analysis.
Authors state that “Our primary endpoint was the achievement of remission at week 12, defined as a CDAI of less than 150 points”. Thus, one would expect a dichotomous outcome based on the week 12 value and a primary analysis of this endpoint. Instead, CDAI is treated as continuous, and comparisons with baseline at 6 and 12 weeks are shown in Table 2 and Figure 1. Please clarify.
Furthermore, some power considerations to support the chosen sample size should be added.
As a second issue, I think the hierarchy of endpoints should be clarified better because the multiplicity adjustment also depends on this. It appears that authors treat secondary endpoints as multiple primary endpoints. In fact, each endpoint is tested at a significant level determined by the method for multiplicity adjustment. Please clarify.
Author Response
Thank you for the review and the comments. Below I am presenting explanation and changes made to the manuscript.
Authors state that “Our primary endpoint was the achievement of remission at week 12, defined as a CDAI of less than 150 points”. Thus, one would expect a dichotomous outcome based on the week 12 value and a primary analysis of this endpoint. Instead, CDAI is treated as continuous, and comparison with baseline at 6 and 12 weeks are shown in Table 2 and Figure 1. Please clarify.
Thank you very much for drawing your attention to this. As our primary endpoint was the achievement of remission at week 12, we transferred the results concerning that to the beginning of the paragraph. We presented the results graphically in Fig.1 and Fig.2 to clarify it.
Furthermore, some power considerations to support the chosen sample size should be added.
We determined post-hoc the power of sample size using G*Power 3.1.9.2 software. For determining that percentage of patients who achieved clinical remission (CDAI < 150) was significantly different from 0, and assuming alpha level = 0.05, in 1st follow-up as well as in 2nd follow-up we achieved the power level (1 minus beta) of 1.0.
We added the missing analysis in Materials and methods section.
As a second issue, I think the hierarchy of endpoints should be clarified better because the multiplicity adjustment also depends on this. It appears that authors treat secondary endpoints as multiple primary endpoints. In fact, each endpoint is tested at a significant level determined by the method for multiplicity adjustment. Please clarify.
We are happy to clarify the way of applying Bonferroni correction for multiple comparisons. This correction was not used between particular endpoints (i.e. taking into account number of endpoints). It was used in post-hoc analysis only and separately for each endpoint. In post-hoc analysis we had for each endpoint 3 comparisons (1st follow-up vs baseline, 2nd follow-up vs. baseline, 2nd vs 1st follow-up) and multiplicity adjustment included those 3 comparisons. We have clarified this in the note under table 2.
Thank you once again for your valuable comments.
Reviewer 2 Report
In the present original article Szczubelek et al showed that, in 32 patients with Crohn’s disease(CD), CD exclusion diet (CDED) was able to achieve clinical remission in the 76.7% after 6 weeks and to lower calprotectin levels. Main comments:
1) The main drawback of this study is that a control group is absent. Without a control group, it impossible to ascertain whether improvement is due to diet or to pharmacological standard of care.
2) Page 1 lines 35-42: discussion about NOD2 is unnecessary for this paper and may be deleted.
3) Were patients allowed to eat fruit or vegetables? (apart from apples and potatoes). Which was the main source of fibres?
4) In order to understand the effectiveness of CDED, it is important not only to report CDAI score before-after, but to elucidate how many patients were in active phase at baseline and how many of them achieved clinical remission after CDED. In this regard, data reported in lines 213-219 would be better represented graphically and compared by a statistical test.
5) Limitations have not been discussed.
Author Response
Thank you for the review and the comments. Below I am presenting explanation and changes made to the manuscript.
1) The main drawback of this study is that a control group is absent. Without a control group, it impossible to ascertain whether improvement is due to diet or to pharmacological standard of care.
Patients undergoing pharmacological treatment were included in the study, when they lost the treatment effectiveness, understood as increase in CDAI>150points. The treatment must have lasted for more than 8 weeks, in fixed doses. In this case, the improvement of the patient's clinical condition, with a stable dose of the drug, was dependent on an additional factor, i.e. on dietary intervention.
To clarify the above, we supplemented the methodology with the missing description (lines: 79-85).
2) Page 1 lines 35-42: discussion about NOD2 is unnecessary for this paper and may be deleted.
NOD2 discussion has been removed.
3) Were patients allowed to eat fruit or vegetables? (apart from apples and potatoes). Which was the main source of fibres?
The CDED allows of up to 18 to 20g of fiber per day. Patients are obligated to eat chicken breast, 2 eggs, 2 fresh potatoes, 2 bananas and 1 apple per day (this is known as “mandatory food”). However, they can also eat food products from a list, known as “allowed foods and beverages”, which have no negative effect on intestinal mucosa. Fruits and vegetables included in the 1st phase are: 1 avocado/day, 5 ripe strawberries/day, 1 slice of melon/day, 2 tomatoes/day, 2 cucumbers/day, 1 carrot/day, 1 cup of fresh spinach, 3 lettuce leaves; in 2nd phase the list is extended by 1 pear, peach or kiwi/day, 10 blueberries/day, 1 large or 2 small zucchini, 4-6 fresh mushrooms, 2 broccoli or cauliflower florets and from week 10 of diet – all fruits and vegetables can be introduced in small quantities. Foods that are not on the list of mandatory or allowed foods are disallowed. The “disallowed foods and beverages” include dried fruits, frozen vegetables, kale, leeks, asparagus, artichoke, celery. These products are known to have a pro-inflammatory effect on intestinal mucosa. All patients received a handout with instructions regarding the diet, including a list of allowed and disallowed foods and products.
In our study the main source of fibre were potatoes, bananas and apples. The food products from “allowed foods and beverages” list were consumed in smaller quantities.
To clarify instructions given to patients, lines 99-104 were added.
4) In order to understand the effectiveness of CDED, it is important not only to report CDAI score before-after, but to elucidate how many patients were in active phase at baseline and how many of them achieved clinical remission after CDED. In this regard, data reported in lines 213-219 would be better represented graphically and compared by a statistical test.
The results concerning number of patients with clinical remission and clinical response were transferred to the beginning of the paragraph and presented graphically in Fig.1 and Fig.2.
100% of patients included in the study were in the active phase of CD as it was the inclusion criterion for participation in the study. After CDED 76.7% of them achieved clinical remission (CDAI<150 points).
The data reported in 213-219 lines were represented graphically.
5) Limitations have not been discussed.
Limitations of the study have been added and discussed in lines 311-321.
Thank you once again for your valuable comments.
Round 2
Reviewer 2 Report
Regarding point 1, Authors misunderstood my comment. A control group should be constituted by patients who consumed a standard diet
Author Response
Thank you for the response. I would like to explain why we could not have a control group with patients who consumed a standard diet.
Leaving patients with active CD on standard diet would mean leaving them untreated, which is ethically unacceptable. Bearing in mind the period of the COVID-19 pandemic, we chose an observational study in which we evaluate the effectiveness of dietary intervention with proven effectiveness in the paediatric population. The next step will be to compare CDED intervention with glucocorticoid therapy.